# Streamlining Acute Stroke Processes and Data Collection: A Narrative Review

**DOI:** 10.3390/healthcare12191920

**Published:** 2024-09-25

**Authors:** Adam Forward, Aymane Sahli, Noreen Kamal

**Affiliations:** 1Department of Industrial Engineering, Faculty of Engineering, Dalhousie University, Halifax, NS B3H 4R2, Canada; adam.forward@dal.ca (A.F.); aymane.sahli@dal.ca (A.S.); 2Department of Community Health and Epidemiology, Faculty of Medicine, Dalhousie University, Halifax, NS B3H 4R2, Canada; 3Department of Medicine (Division of Neurology), Faculty of Medicine, Dalhousie University, Halifax, NS B3H 4R2, Canada

**Keywords:** acute stroke, workflow standardization, resource management, data collection

## Abstract

(1) Background: Acute ischemic stroke treatment has been thoroughly studied to identify strategies to reduce treatment times. However, many centers still struggle to achieve fast treatment times. Additionally, studies primarily focus on larger, more advanced centers; yet, smaller centers often face longer treatment times. (2) Objectives: The aim of this study is to analyze the existing literature reviewing stroke treatment processes in primary and comprehensive stroke centers that investigated or reduced treatment times. The articles identified were categorized based on the focus areas and approaches used. (3) Results: Three main categories of improvements were identified in the literature: (1) standardization of processes, (2) resource management, and (3) data collection. Both primary and comprehensive stroke centers were able to reduce treatment times through standardization of the processes. However, challenges such as variations in hospital resources and difficulties incorporating data collection software into workflow were highlighted. Additionally, many strategies to optimize resources and data collection that can benefit primary stroke centers were only conducted in comprehensive stroke centers. (4) Conclusions: Many existing strategies to improve stroke treatment times, such as pre-notification and mass stroke team alerts, have been implemented in both primary and comprehensive stroke centers. However, tools such as simulation training are understudied in primary stroke centers and should be analyzed. Additionally, while data collection and feedback are recognized as crucial for process improvement, challenges persist in integrating consistent data collection methods into clinical workflow. Further development of easy-to-use software tailored to clinician needs can help improve stroke center capabilities to provide feedback and improve treatment processes.

## 1. Introduction

Stroke is one of the leading causes of death and disability globally [1]. Ischemic stroke accounts for approximately 85% of all strokes and is treatable with two methods: thrombolysis [2,3] and endovascular thrombectomy (EVT) [4,5]. Both treatments are highly time-dependent, as approximately 1.9 million neurons die in the brain every minute a stroke goes untreated [6]. Thrombolysis and EVT must be provided within 4.5 h and up to 24 h from symptom onset, respectively. Both treatments are more effective the earlier they are administered [5,7].

Benchmark times have been in place for hospitals to measure their performance. The first benchmark time is from arrival to the start of obtaining the CT scan within 15 min, which is door-to-imaging time (DTI). The second benchmark is the time from arrival to the administration of thrombolysis within a median 30 min, which is door-to-needle time (DNT), and the third benchmark is to achieve a median time from arrival to the start of EVT within 60 min, which is door-to-groin puncture time (DGPT) [8,9]. Studies have focused on strategies primarily to reduce DTI and DNT and have found out that a DNT of 20 min was achievable [10]. Additional studies have identified and evaluated the effects of individual strategies to improve the stroke processes of care [11,12,13], primarily focusing on DNT. These studies highlighted the importance of organized practice and continuous improvement.

Rural and community hospitals are typically primary stroke centers (PSCs) and are capable of thrombolysis treatment; urban teaching centers are called comprehensive stroke centers (CSCs) and are capable of both thrombolysis and EVT. Despite the proposed strategies to improve DNT, many centers still do not meet the median benchmark time of 30 min [14], reducing both the number of patients that may be eligible for thrombolysis and the expected effectiveness of the treatment [5,7]. Patients that first arrive at a PSC often face longer treatment times, as these centers often do not have in-house neurology teams, relying on their emergency physicians to treat strokes, while CSCs have more advanced equipment and dedicated neurology teams. Additionally, patients that first arrive at a PSC but are eligible for EVT must be transferred to the nearest CSC, increasing the overall time to receive EVT. Despite these challenges, most studies focusing on DNT are conducted in CSCs [15,16].

This narrative review aims to compare strategies implemented in PSCs and CSCs to review the barriers different centers face to optimizing and monitoring their stroke treatment processes. This review aims to advance the current knowledge by investigating similarities and differences in the capabilities of PSCs and CSCs in treating stroke and to review whether PSCs suffer any disparities in improving stroke treatment practices based on their capabilities. It also aims to compare and categorize the approaches taken by PSCs and CSCs based on the strategies used to investigate and improve their processes.

## 2. Search Strategy

The review was conducted using PubMed to identify studies improving acute stroke treatment. The keywords used included: “acute stroke”, “process”, and “workflow” (door-to or door to). The words “rehabilitation” and “transfer” were excluded from the search to focus on interventions that were applicable to the in-hospital workflow of both PSCs and CSCs. The articles included were: (1) published in peer-reviewed journals; (2) written in English; (3) published within the last 10 years; and (4) provided a set of approaches to either identify or improve treatment processes either in single centers or across multiple centers.

Two hundred and eighty-six records were identified from the keyword search. Titles and abstracts were screened for relevance, and observational studies that examined the in-hospital processes of a single site over time or compared processes between multiple sites were selected. The papers included were summarized and categorized based on their focus areas and approaches used, including process maps, driver diagrams, standard operating procedures, failure modes, effects, and criticality analysis, etc. 

## 3. Categorized Strategies

The literature identified three main categories of improvement strategies in different centers. The first was the standardization of processes. This concerned how the centers organized stroke care and implemented clinical practice guidelines to optimize their stroke care pathway. The second was resource management, which involved how the dedicated stroke teams distributed tasks between their members to deliver care effectively. Finally, data collection focused on how the centers gathered and used data to either guide improvements or provide feedback to influence change. Table 1 shows the included articles and their focus areas.

### 3.1. Standardization of Processes

The most frequently implemented strategies in single centers involved standardizing workflow processes. The implementation of standardization discussed four main methods to reduce treatment times: task completion pre-arrival, streamlining workflow, facilitating communication, and parallel task flow. 

Task completion pre-arrival was identified as a key factor in reducing treatment times as it allowed the stroke team to prepare before the patient arrival at the hospital. If pre-notification occurs, further steps, including pre-registering patients, paramedics putting in IV lines, pre-ordering lab tests, and a CT scan, can be completed. McGrath et al. (2018) discussed not having the capabilities to pre-register patients in their center (a PSC) due to their hospital’s electronic system, and this may have prevented their center from reducing DNTs further [33]. In addition, Mohedano et al. (2017) identified how often pre-arrival tasks were not completed and what caused the failures [35].

Streamlining workflow was completed by modifying the centers’ existing protocols. Often, modifying the protocols would require a multidisciplinary team to meet and agree on the planned changes to the process. Frequently identified strategies to streamline the process included direct to CT (or emergency department (ED) ‘pit stop’ evaluations), ensuring the stroke team is involved in treatment, beginning the thrombolysis treatment discussion as soon as imaging ends, and speeding up the thrombolysis treatment by preparing and administering the drug in the imaging area. Regardless of whether it was a PSC or CSC, many centers implemented the described strategies. Kamal et al. (2017) identified the frequency and time saved by iteratively implementing rapid registration, CT, and thrombolysis administration practices, showing mean time reductions ranging from 7 min to 19 min [11]. Kamal et al. (2019) and Hennebry et al. (2022) implemented interventions to reduce DNT in different PSCs to 30 min and 35 min, respectively [30,39]. Two studies from different center types (one CSC and one PSC) did not discuss initiating thrombolysis in the imaging area [25,27], and this strategy could be more difficult for PSCs without neurology teams to implement; however, comfort with administering thrombolysis in the scanner increased with simulation training [19]. 

Facilitating communication was an improvement strategy that primarily relied on simple methods of providing information to multiple people at once and was applied to both PSCs and CSCs. Zuckermann et al. (2016), Hennebry et al. (2022), and Hill et al. (2020) modified their communication systems to alert all stakeholders simultaneously, rather than individually, and identified this strategy as a key factor in reducing their DNTs [22,27,30]. This not only applies to pre-notification, but also intra-departmental communication, as one study applied the use of a “mass alert” to notify the ED staff when the EVT decision was made [22]. 

Parallel task flow was a strategy implemented in both PSCs and CSCs. However, CSCs often had larger stroke teams, including in-house neurologists, residents, fellows, and stroke nurses. The use of a stroke nurse aided in setup and administration of thrombolysis in the CT scanner or imaging area in some CSCs [17]. Additional studies included the ED social worker and pharmacist, but this was only conducted in CSCs [20,29]. When modifying protocols, centers would assign roles and responsibilities to each member of the stroke team, providing each member with clarification of their roles. In PSCs, the smaller stroke teams meant that tasks such as patient history review, clinical assessments, and contacting family members were the responsibility of the nurse or physician [30].

### 3.2. Resource Management

One of the challenges discussed regarding PSCs is the lack of resources, primarily in terms of staff, equipment, and training [15,37]. Resource variations can occur based on the location, but also the time of day. Buleu et al. (2024) found that DTI times increased in their center during night shift hours, but also found that thrombolysis rates increased during night shift hours [44]. Common strategies have been discussed to improve these challenges in PSCs, including the use of telestroke [46,47,48]. However, variability in neurology contact time can impact the effectiveness of telestroke implementation [49]. Besides telestroke, other strategies can be used to aid PSCs in maximizing resource efficiency.

Resource management problems occur when there is inadequate stroke team size or staff specialization. In PSCs, the roles involved in stroke treatment are typically less specialized. Some PSCs have neurology teams in-house, while others rely on telestroke. Bulmer et al. (2021) interviewed rural and urban clinicians and found that rural clinicians often relied on telestroke consultations for treatment decisions, and there could be delays contacting the neurologist [43]. Prabhakaran et al. (2015) compared an academic and community hospital and found that the community hospital faced unique challenges with ED overcrowding [18]. Liberman et al. (2023) identified that the most critical failures that occurred in PSCs and CSCs were due to providers lacking experience in assessing patients or ED busyness resulting in missed paramedic reports [23]. While studies have discussed the lack of staff as being a contributor to challenges in stroke treatment in PSCs, Wong et al. (2023) reviewed stroke treatments in their PSC and found that the number of staff present did not impact the team journey time [38]. 

Along with modifying protocols, PSCs without neurologists accommodated their lack of specialized teams by relying on decision support references. The references would be used to aid in the process that relies on algorithms to determine the decisions the clinician would have to take. This included determination of thrombolysis and EVT referral, exclusion from advanced imaging, and calculating scores like ASPECTS. However, Hennebry et al. (2022) discussed including a video guide for staff education and training in future process improvement initiatives [30]. The decision support references were only described as an intervention implemented in PSCs without in-house neurology teams [30,33].

Finally, a strategy that can aid in resource management is the use of simulation training to improve staff experience. Willems et al. (2019) explain the importance of non-technical skills, such as communication, as necessary skills to develop for stroke treatment, and regular simulation training can aid in reducing treatment times [24]. While the two studies discussing simulation training focused on CSCs, Tahtali et al. (2017) explained how to setup a stroke team simulation and how it can be applied in different centers [42].

Despite the challenges associated with fewer resources in PSCs, existing strategies can be implemented to address the lack of specialization or potential unfamiliarity with stroke processes, including the regular use of telestroke, decision support tools, and simulation training. 

### 3.3. Data Collection

The final set of strategies used to facilitate improvements was the collection and use of data gathered in the process. There were two aspects of data collection reviewed: how the data were collected and how they were used. Studies have identified and implemented data feedback as an improvement strategy for DNT [14,30,35]. However, data feedback was less discussed than the previous research and was only the focus of eight studies.

Jauch et al. (2018) evaluated the treatment times in a set of thrombolysis centers called the “stroke belt” and used retrospective electronic medical record (EMR) reviews to determine each hospital’s DNT. The data from the chart review were leveraged by comparing the clinicians’ anticipated DNTs with their actual DNTs, highlighting that their actual DNTs were much longer [34]. This study relied on education and data feedback to motivate clinicians to improve their DNTs, and their rates of patients receiving thrombolysis within 60 min increased from 22.2% to 67.3% [34]. 

Multiple studies discussed prospective (or real-time) data collection and feedback methods. A simple real-time data feedback mechanism was implemented by Burnett et al. (2014) using a group text system [31]. Through regular real-time feedback after each stroke treatment, documenting the DNTs and reviewing cases that were longer than 60 min, the intervention reduced the centers’ DNTs from 82 min to 56 min. Despite the simplicity, the prompt data feedback was beneficial in motivating change.

One of the studies focused on real-time data collection used sensors in key locations (ED, imaging, and angiosuite) to collect time metrics throughout the treatment process. Zhang et al. (2023) combined EMR information and sensor data to identify when delays were occurring and relay feedback to the stroke team [26]. The benchmark times improved significantly after the intervention, but this intervention was only implemented in a large CSC that had a specialist abstracting the data to the medical charts.

One of the approaches to facilitate prospective data collection was through mobile software. Three studies focused on the use of mobile applications that aimed to aid in treatment processes while facilitating documentation. The studies focused on three different applications: Pulsara’s “Stop Stroke” [21], the Mayo Clinic Acute Stroke Evaluation tool [36], and JOIN [28]. Each software platform served similar functions and provided aids to assist clinicians. Each software platform provided a combination of communication tools, reference aids, and timestamp capabilities. There were key differences in the software’s features. Pulsara’s “Stop Stroke” used the phone notification system to update the stroke team on tasks completed [21]; the Mayo clinic Acute Stroke Evaluation software provided an automatic text feature to contact the on-call neurologist if the center used telestroke [36]; and JOIN linked to imaging software to provide automatic updates on imaging results [28]. Each study found that the software helped to reduce treatment times. However, there were reported issues from each study concerning either data completeness [21,36], interference with workflow [28], or low user satisfaction [28]. A final software platform called “Code Stroke Alert” incorporates unique features, including geolocation and tiered notifications to communicate relevant information to staff with different roles [45]. However, pilot test results have not yet been published for the software.

## 4. Discussion

Many improvement strategies have been implemented in the stroke treatment process across PSCs and CSCs. Despite limitations in resources, some PSCs can reduce their treatment times significantly using similar strategies to those of CSCs. However, due to the smaller stroke team sizes and less specialized staff, PSCs without neurologists have implemented additional aids, such as referral guides and regular feedback, to improve performance [30,33]. Many of the strategies implemented should be applicable to both center types; however, the literature focuses more on CSCs than PSCs; so, some approaches to reducing treatment times (such as simulation training) need to be further researched to analyze their viability in PSCs and to ensure they are generalizable. While barriers such as costs and technology systems can prevent the implementation of certain improvements in PSCs, many interventions were cost-neutral and did not require additional resources to implement.

While standardization of tasks has been thoroughly reviewed and implemented in stroke process improvement in both PSCs and CSCs, resource management strategies were less discussed. Telestroke is often seen as the primary method to improve PSC performance through neurologist consultations, but reference guides and simulation training are less discussed. Reference guides were important interventions discussed for PSCs to improve clinicians’ confidence in clinical decision making. Future studies should consider the integration of clinical decision support tools in PSCs to determine their effectiveness.

Simulation training was shown to be valuable in improving communication and confidence in staff in CSCs [19,24]. With evolving technology, simulation training can become more accessible through technology such as virtual reality [50]. However, an important aspect highlighted in the studies includes communication among the stroke team members; so, group simulation training should not be ignored. Future research should incorporate simulation training in PSCs, as the effects of simulation training improved communication skills, familiarity with stroke protocols, and comfort treating with thrombolytics, which are critical for PSCs with less specialized personnel.

Data collection and monitoring was also discussed in fewer studies, despite its recognized value in improving treatment processes. Prompt feedback was shown to reduce treatment times and improve motivation in the centers. The studies in PSCs relied on retrospective data collection that required the dedicated quality improvement intervention team to complete. Existing tools were tested to aid in prospective data collection, but they suffered from issues that were due to missing data, intensive requirement of resources, or interference with workflow. Studies discussing mobile application use found that users were missing key data, or they were not satisfied with the software [21,26,28,36]. Further research is needed to identify the critical needs to improve the ease of data collection and feedback mechanisms, improving the ability of PSCs to consistently utilize data to understand their process and areas of improvement.

This review study was limited to various aspects. First, topics including improvement strategies in transferring patients from PSCs to CSCs and user experience analyses in acute care settings could be further reviewed to identify other applications. Second, this study did not consider the hospitals’ geographic locations, and further studies could investigate and compare stroke processes and improvements in different areas, such as developing countries and countries with public or private healthcare systems. Third, despite careful review, some relevant articles could have been inadvertently excluded. Finally, this review analyzed the current treatment process according to three different aspects. However, the stroke treatment processes in both centers can be affected by other aspects as well, such as comfort level of treating and hospital culture. 

## 5. Conclusions

There are many accepted strategies to reduce DNT by standardizing stroke processes. Streamlining workflow, completing tasks before the patient arrives, simplifying communication, and completing tasks in parallel are well-established strategies that both PSCs and CSCs can utilize to improve their processes. While PSCs face unique challenges concerning their resource availability, multiple centers have effectively reduced their DNT through task standardization and the use of tools like telestroke. However, existing strategies, such as decision support tools and simulation training, have shown to be effective in CSCs and are understudied in PSCs. Furthermore, data feedback, while shown to be an effective method to reduce treatment times, has few dedicated strategies in place to effectively collect and relay in PSCs and many CSCs. Thus, these existing strategies about resource management and data collection should be implemented in PSCs to improve their processes.

## Figures and Tables

**Table 1 healthcare-12-01920-t001:** Focus areas and summary of results of included articles.

Authors	Article Focus					
	Hospital Focus (PSC, CSC, Both)	Standardization of Processes	Resource Management	Data Collection	Summary of Strategies/Study	Outcomes of Study
Ernst et al. (2022) [17]	CSC	Yes	No	No	Pre-notification; having the physician in the ED pre-arrival; pre-registration, direct to CT policy; early treatment decision making; emergency consent procedures if no witness is available; hired a stroke nurse; added thrombolysis kit.	Reduced DNT from 51 to 29 min.
Prabhakaran et al. (2015) [18]	Both	No	Yes	No	Identified time delay failures in a PSC and CSC and unique challenges in the PSC concerning overcrowding, waiting for registration, and delays obtaining consent. The CSC had failures concerning delays obtaining imaging without registration and delays activating the stroke team.	Identified unique treatment delays occurring in PSCs compared to CSCs.
Bohmann et al. (2022) [19]	CSC	Yes	No	No	In situ simulation training; increased involvement with EMS; increased thrombolysis rates in the CT scanner.	Reduced DNT in simulation-experienced teams from 39 to 33 min and DGPT from 95 to 74 min.
Aghaebrahim et al. (2019) [20]	CSC	Yes	Yes	No	Pre-notification; physician in ED pre-arrival, direct transport to CT scanner, added neurologist during off-hours.	Reduced DNT from 56 to 30 min.
Andrew et al. (2017) [21]	Both	No	No	Yes	Identified factors affecting DNT through data collected in Stop Stroke; found earlier activation by EMS providers was associated with shorter DNT.	Identified factors impacting DNT and showed the use of Stop Stroke was correlated with a 7 min DNT reduction.
Hill et al. (2020) [22]	CSC	Yes	No	No	Modified protocol based on night shift hours; implemented mass alert system of treatment decision; included ED social worker into workflow to contact a witness.	Reduced DGPT from 110 to 80 min.
Liberman et al. (2022) [23]	Both	Yes	No	No	Identified delays in centers’ stroke process. including lack of symptom recognition, incomplete medical histories, and non-recognition of “walk-in” symptoms”; suggested solutions including communication tools for information sharing, redesign of the stroke team to formalize roles and responsibilities, and the creation of an automated best-practice alert for use in the ED.	Highlighted major delays that occurred in the stroke treatment process and proposed solutions to reduce severity of delays.
Willems et al. (2019) [24]	CSC	Yes	Yes	No	In situ simulation training; thrombolysis kits; team training; clear task allocation; pocket card reference guides; standard documentation forms.	Achieved a mean DNT under 25 min and increased thrombolysis rates to 30%.
Park et al. (2022) [25]	PSC	Yes	Yes	No	Pre-notification; IV access pre-arrival; hired a stroke nurse; direct to CT protocol; pre-registration; ‘code stroke’ page system; added staff hours; increased CT availability.	Reduced median DNT from 70 to 51 min.
Zhang et al. (2023) [26]	CSC	No	No	Yes	Used radiofrequency sensors to timestamp stroke processes in the study center (a large CSC). Identified early treatment decision communication, unobstructed aisles, group alert systems, and improved stroke knowledge and education as areas of improvement.	Reduced DNT from 118 to 26 min.
Zuckerman et al. (2016) [27]	CSC	Yes	Yes	No	Notified the Interventional Radiology team earlier for higher NIHSS scores; mass alert paging system; completion of NIHSS in the CT scanner; clear task allocation.	Reduced DNT from 63 to 44 min.
Kamal et al. (2017) [11]	CSC	Yes	No	No	Added triage level for severe stroke; registering patients as unknown; administering thrombolysis in scanner; direct to CT protocol on EMS stretcher.	Reduced DNT from 53 to 35 min and determined the % reduction in DNT for individual interventions.
Gutiérrez-Zúñiga et al. (2022) [28]	CSC	Yes	No	Yes	Used the JOIN software to facilitate information sharing through mobile devices and collected timestamps for process benchmarks; found the JOIN-activated strokes had shorter DNT but found user satisfaction was hindered by the added work using the software.	Reduced DNT from 51 to 36 min for clinicians using the JOIN software.
Busby et al. (2016) [29]	CSC	Yes	No	No	Pre-notification; mass alert stroke team; direct to CT protocol on EMS stretcher; added the pharmacist for mixing and delivery of thrombolysis in the scanner.	Reduced DNT from 62 to 25 min.
Hennebry et al. (2022) [30]	PSC	No	Yes	No	Pre-alert system for stroke team; FAST registration; ED ‘pit stop’ evaluation; modified CT ordering codes; added reference EVT guidelines; added thrombolysis kit.	Reduced DNT from 105 to 35 min in first 9 months.
Burnett et al. (2014) [31]	CSC	Yes	No	Yes	Added a text message system to share DNT results to stroke team after cases; the team reviewed DNTs over one hour within 24–72 h and conducted bi-weekly dashboard updates.	Reduced DNT from 82 to 56 min.
Van Schaik et al. (2014) [32]	PSC	Yes	No	No	Formed a stroke team; added pre-notification; added INR device used on suspected anticoagulant cases; added policy for CT priority for stroke; administered thrombolysis on CT table.	Reduced DNT from 60 to 25 min.
McGrath et al. (2018) [33]	PSC	Yes	Yes	No	Added a stroke team; included treatment reference guides in process for inexperienced staff; added policy for CT priority for stroke.	Reduced DNT from 99 to 67 min.
Jauch et al. (2018) [34]	PSC	Yes	No	Yes	Conducted surveys to compare perceived versus actual treatment performance; surveyed clinicians on barriers and challenges to identify areas for improvement; completed educational sessions; collected and disseminated data to sites to measure progress.	Increased rate of patients receiving thrombolysis within 60 min from 1.9% to 5.2%.
Mohedano et al. (2017) [35]	CSC	Yes	No	No	Reviewed history and ordered CT pre-arrival; INR device used on suspected anticoagulant cases; direct to CT protocol; making the thrombolysis decision immediately after CT.	Reduced DNT from 52 to 40 min. Identified causes of non-compliance with best practices.
Rubin et al. (2015) [36]	CSC	Yes	Yes	Yes	Used mobile software (Mayo Clinic) to provide reference guides, workflow checklists, reminders, and communication features during stroke treatment.	Reduced mean DNT by 16 min.
Prior et al. (2016) [37]	PSC	No	Yes	No	Identified primary challenges associated with rural Australian stroke centers, including costs, retaining staff, lack of training opportunities, access to medical equipment and expertise. Possible solutions discussed included access to telemedicine, rural training opportunities, and hub and spoke models.	Investigated barriers in rural stroke centers and proposed potential solutions to reduce disparities.
Wong et al. (2023) [38]	PSC	Yes	Yes	No	Analyzed workflow of stroke treatment process in their center with body cameras; identified delay factors, including offloading the patient to a bed before CT, not having an IV pre-arrival, and late team arrival times during night hours.	Identified steps in process that increased team journey time and found the center could have an ideal DNT of 21 min.
Kamal et al. (2019) [39]	PSC	Yes	No	Yes	Paramedics establish IV line pre-arrival; single-call activation of stroke team; ED swarm on arrival; small rewards for fast treatments; taking the patient to CT on the paramedics stretcher.	Reduced DNT from 88 to 30 min.
Kircher et al. (2016) [40]	CSC	No	Yes	No	Reviewed pre-hospital and intrahospital strategies like education for stroke symptom recognition, direct to CT protocols, constant neurology team presence, collaboration with EMS, telemedicine in PSCs, and early neurointerventional involvement for suspected large vessel occlusions.	Investigates opportunities in optimizing pre- and intra-hospital processes through ambulance routing, telemedicine, and screening tools.
Santana Baskar et al. (2021) [41]	Both	Yes	Yes	No	Reviewed treatment time reductions in pre-hospital and intra-hospital workflow, reviewed strategies, including pre-notification, parallel workflow, imaging modalities used, thrombolysis kits, and physician feedback.	Investigates opportunities for intra-hospital stroke treatment, highlighting factors affecting implementation of best practices.
Tahtali et al. (2017) [42]	Both	Yes	Yes	No	Provides an overview and guide for setting up a stroke team, steps in the stroke protocol, and steps to complete the stroke simulation training and gather feedback from the stroke team.	Provided guide to implement simulation training, highlighted individual results, increasing rate of patients receiving thrombolysis within 30 min from 21% to 77%.
Bulmer et al. (2021) [43]	Both	Yes	Yes	No	Interviewed clinicians in PSCs and a CSC to compare comfort treating strokes, existing delays, patient consent for treatment, suggested improvements, and mapped thrombolysis process in each center.	Identified differences between rural and urban centers in Nova Scotia, Canada.
Buleu et al. (2024) [44]	PSC	Yes	Yes	No	Conducted an observational study for code strokes during daytime and night hours. Evaluated differences in number of patients arriving, modes of transport, benchmark times, and treatment rates.	Identified variations in DTI times during night shift hours while following the “code stroke alert” protocol.
Seah et al. (2019) [45]	-	No	No	Yes	Provides overview of “Code Stroke Alert” software as a communication and data sharing platform for stroke treatment.	Highlights flow of information sharing, user interface, and features implemented, including tiered notifications and geolocation.

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
