# Peer review of "Streamlining Acute Stroke Processes and Data Collection: A Narrative Review"

_healthcare, 2024, doi:10.3390/healthcare12191920_

Round 1

Reviewer 1 Report

Comments and Suggestions for Authors

In the introduction, all-time targets should be discussed, not just DNT

It would be better if there were no studies older than ten years included in the review (table 1)

You should add more data on statistics

Abbreviations must be separated with commas, next to each other, not one below the other.

Limitations of the study need to be improved

What type of doctors have PSC vs CSC? 

It would be helpful to clarify the difference in the experience of stroke patients between PSC and CSC. This would provide a clear comparison and enhance the reader's understanding.

Discussions need to be improved

To improve the manuscript, the following articles should be cited:

a) doi: 10.3389/fneur.2019.00725

b) https://doi.org/10.3390/jpm14060596

Comments on the Quality of English Language

Minor English editing.

Author Response

1. Summary

Thank you very much for taking the time to review this manuscript. Please find the detailed responses below and the corresponding revisions/corrections highlighted and in track changes in the re-submitted files.

2. Point-by-point response to Comments and Suggestions for Authors

Comment 1: In the introduction, all-time targets should be discussed, not just DNT

Response 1: Thank you for pointing this out. We have included explanations of the main time targets including door-to-imaging time and door-to-groin-puncture time on page 3, paragraph 2.

“Benchmark times have been in place for hospitals to measure their performance. The first benchmark time is from arrival to the start of CT scanning within 15 minutes, which is door-to-imaging time (DTI). The second benchmark is the time from arrival to the to administration of thrombolysis within a median 30 minutes for door-to-needle time (DNT), and the third benchmark is to achieve a median time from arrival to the start of EVT within 60 minutes, which is door-to-groin-puncture (DGPT).”

Comment 2: It would be better if there were no studies older than ten years included in the review (table 1)

Response 2: Thank you for pointing this out. On review, two articles were considered that were older than ten years, Ford et al (2012) and Olson et al. (2011). As suggested, these two studies have been removed.

Comment 3: You should add more data on statistics

Response 3: Thank you for the suggestion. We have included additional columns to Table 1 to highlight the results from the studies, including the reduction in treatment times, increase in treatment rates, or significant findings. We hope including this information gives a clearer view of the impact of the studies.

Comment 4: Abbreviations must be separated with commas, next to each other, not one below the other.

Response 4: Thank you for pointing this out. We have adjusted the abbreviations to separate them with commas in section 7.

Comment 5: Limitations of the study need to be improved

Response 5: Thank you for the feedback. We have included additional limitations we found that were of relevance including not considering geographic location or the healthcare systems of different countries. We also include that there is potential some relevant articles could have been excluded despite careful review. The changes are in paragraph 2 of page 17:

“This review study was limited to various aspects. First, topics including improvement strategies in transferring patients from PSCs to CSCs and user experience analyses in acute care settings could be further reviewed to identify other applications. Second, this study did not consider the hospitals geographic locations, and further studies could investigate and compare stroke processes and improvements in different areas such as developing countries and countries with public or private healthcare systems. Third, despite careful review, some relevant articles could have been inadvertently excluded. Finally, this review analyzed the current treatment process in three different aspects. However, stroke treatment process in both centers can be affected in other aspects as well, such as comfort level of treating and hospital culture.”

Comment 6: What type of doctors have PSC vs CSC? 

Response 6: We have clarified that PSCs often rely on emergency physicians to treat stroke patients, but some PSCs can have in-house neurology teams as stated in paragraph 3 of page 3

“Patients that first arrive to a PSCs often face longer treatment times, as these centers often do not have in-house neurology teams, relying on their emergency physicians to treat strokes, while CSCs have more advanced equipment and dedicated neurology teams.”

Comment 7: It would be helpful to clarify the difference in the experience of stroke patients between PSC and CSC. This would provide a clear comparison and enhance the reader's understanding.

Repsonse 7: Thank you for your suggestion. We have clarified the unique experience for stroke patients in PSCs in paragraph 3 of page 3 as follows:

“Patients that first arrive to a PSCs often face longer treatment times, as these centers often do not have in-house neurology teams, relying on their emergency physicians to treat strokes, while CSCs have more advanced equipment and dedicated neurology teams. Additionally, patients that first arrive to a PSC but are eligible for EVT must be transferred to the nearest CSC, increasing the overall time to receiving EVT”

Comment 8: Discussions need to be improved

Repsonse 8: We have rewritten sections of the discussion to elaborate on recommendations that resulted from the narrative review.

Comment 9: To improve the manuscript, the following articles should be cited:

a) doi: 10.3389/fneur.2019.00725

b) https://doi.org/10.3390/jpm14060596

Response 9: Thank you for your suggestion. We have included the following articles to further discuss how variations can occur between day and night shifts in centers with established “code strokes” and the use of mobile software to facilitate data collection and communication.

4. Response to Comments on the Quality of English Language

Point 1: Minor English editing.

Response 1: We have made edits to improve the readability of the text

Reviewer 2 Report

Comments and Suggestions for Authors

in attachment

Author Response

1. Summary

2. Point-by-point response to Comments and Suggestions for Authors

Comment  1. Abstract Clarity and Focus:

 The abstract provides a good overview, but it could be more concise. Consider revising sentences to improve readability, especially the sentence on line 17 that is slightly convoluted. A clearer distinction between the key findings and implications for practice would be beneficial.

Response 1: Thank you for pointing this out. We have revised sentences to improve readability and more clearly highlighted the key findings concerning the underuse of simulation training in PSCs and required development of data collection mechanisms to improve data collection and feedback for stroke treatment in both PSCs and CSCs.

Comment 2: Introduction and Background: The introduction is thorough but could benefit from a stronger emphasis on the gap this review addresses. While the rationale for focusing on smaller centers and their challenges is mentioned, it would be helpful to explicitly state how this review advances current knowledge.

Response 2: Thank you for pointing this out. We have added additional emphasis on addressing the gap concerning reviewing the stroke treatment capabilities of PSCs and CSCs and how we aim to identify any disparities in how both types of centers are able to streamline their processes.

“This review aims to advance the current knowledge by investigating similarities and differences in the capabilities of PSCs and CSCs in treating stroke and review if PSCs suffer any disparities in improving stroke treatment practices based on their capabilities”

Comment 3:  Line 51-54: The description of challenges faced by PSCs could be expanded with specific examples of how these challenges impact patient outcomes.

Response 3: Thank you for the suggestion. We have specified that the increased time to treatment due to challenges in PSCs reduces the potential for stroke patients to be treated and the effectiveness of treatment as a result of the time delay in paragraph 3 (pg. 3)

“Despite the proposed strategies to improve DNT, many centers still do not meet the median benchmark time of 30 minutes, reducing both the number of patients that may be eligible for thrombolysis, and the expected effectiveness of the treatment.”

Comment 4:  The methodology section is clear but could include more detail regarding the selection criteria for the articles reviewed.

Response 4: Thank you for pointing this out. We have added an additional paragraph specifying how articles were screened on if they completed observational studies examining hospital processes, or compared processes between multiple centers.

“286 records were identified from the keyword search. Titles and abstracts were screened for relevance, and observational studies that examined in-hospital processes of a single site over time or compared processes between multiple sites were selected. Papers included were summarized and categorized based on their focus areas and approaches used, including process maps, driver diagrams, standard operating procedures, failure modes, effects, and criticality analysis, etc.”

Comment 5:  Line 60-66: The search strategy is outlined, but the justification for the chosen time frame (post-2010) and databases could be more explicit. Additionally, consider including a flow diagram to visualize the article selection process.

Response 5: Thank you for the feedback. It was suggested that we only include articles that were within the past 10 years, which we have done so. We have added further explanation on the article screening process which we believe provides a comprehensive explanation of the article selection process including abstract screening and full text screening in page 4

“286 records were identified from the keyword search. Titles and abstracts were screened for relevance, and observational studies that examined in-hospital processes of a single site over time or compared processes between multiple sites were selected. Papers included were summarized and categorized based on their focus areas and approaches used, including process maps, driver diagrams, standard operating procedures, failure modes, effects, and criticality analysis, etc.”

Comment 6:  Results and Categorization of Strategies: The categorization of strategies into standardization of processes resource management, and data collection is logical. However, the manuscript would benefit from a more critical analysis of the effectiveness of these strategies, especially concerning their applicability to different types of stroke centers.

Response 6: Thank you for your suggestion. Many of the strategies implemented were feasible and implemented in both PSCs and CSCs. We have added a sentence explaining that the main strategy that PSCs may not be able to incorporate is the administration of thrombolysis in the CT scanner, as this was often the stroke nurses responsibility in CSCs. However, the remaining strategies were not indicated to have barriers towards implementation.

Comment 7:  Table 1 (Line 74): The table is informative, but it could be improved by adding a column that briefly summarizes the outcomes or findings from each study, providing readers with a quick reference to the effectiveness of the strategies discussed.

Repsonse 7: Thank you for your suggestion. We have added two additional columns to the table. The first summarizing the study, then a second summarizing the key results of the study to provide a comprehensive overview of the studies reviewed.

Comment 8: 5. Discussion: The discussion is comprehensive but could be more balanced by including limitations of the studies reviewed. For example, addressing the potential biases in the literature (e.g., more studies focusing on CSCs) would strengthen the review.

Repsonse 8: Thank you for the suggestion. We have included a section in the discussion highlighting that more research is needed to determine if the strategies implemented in CSCs are generalizable to PSCs.

“Many of the strategies implemented should be applicable to both center types, however, the literature focuses more on CSCs than PSCs, so some approaches to reduce treatment times (such as simulation training) need to be further researched to analyze their viability in PSCs and ensure they are generalizable.”

Comment 9:  Line 202-207: While the discussion on resource management is insightful, it would be beneficial to include potential solutions or recommendations for overcoming the challenges identified, particularly in PSCs. It could be enhanced by including a more explicit call to action or recommendations for future research. This would provide a clearer takeaway for practitioners and researchers.

Response 9: Thank you for your suggestion. We have added more explicit recommendations for future research to conduct simulation training in PSCs as discussed in paragraph 2 of the discussion section (pg. 16)

“Future research should incorporate simulation training in PSCs, as the effects of simulation training improved communication skills, familiarity with stroke protocols, and comfort treating with thrombolytics, which are critical for PSCs with less specialized personnel.”

4. Response to Comments on the Quality of English Language

Point 1:  Quality of Writing: The manuscript is generally well-written, but there are occasional grammatical errors and awkward phrasings. For example, the phrase "many centers still do not meet the benchmark time metrics" (line 13) could be revised for clarity.

Response 1: Thank you for your suggestion. We have edited the manuscript to improve phrasing.

Point 2:  Ensure consistency in the use of abbreviations. For instance, the abbreviation for door-to-groin puncture time (DGPT) is introduced on line 251, but it should be defined earlier when first mentioned. (Line 43 is when DGPT is first mentioned)

Response 2: Thank you for pointing this out. We have reviewed the abbreviations used to ensure they are used consistently.

Point 3: Consider including a figure or diagram that visualizes the main pathways or strategies for improving stroke treatment times, which could serve as a helpful summary for readers

Response 3: Thank you for your suggestion. Since we have updated the table to summarize the strategies implemented in different centers alongside the outcomes of the studies, we believe this will provide sufficient details for readers to understand the identified strategies.

Round 2

Reviewer 1 Report

Comments and Suggestions for Authors

The manuscript has improved. The article can be published. Good Luck

Comments on the Quality of English Language

English Language is ok